# Unsupervised Discovery of Parts, Structure, and Dynamics

**Zhenjia Xu**[*]
MIT CSAIL, Shanghai Jiao Tong University

**Zhijian Liu**[*]
MIT CSAIL

**Chen Sun**
Google Research

**Kevin Murphy**
Google Research

**William T. Freeman**
MIT CSAIL, Google Research

**Joshua B. Tenenbaum**
MIT CSAIL

**Jiajun Wu**
MIT CSAIL

## Abstract

Humans easily recognize object parts and their hierarchical structure by watching how they move; they can then predict how each part moves in the future. In this paper, we propose a novel formulation that simultaneously learns a hierarchical, disentangled object representation and a dynamics model for object parts from unlabeled videos. Our Parts, Structure, and Dynamics (PSD) model learns to, first, recognize the object parts via a layered image representation; second, predict hierarchy via a *structural descriptor* that composes low-level concepts into a hierarchical structure; and third, model the system dynamics by predicting the future. Experiments on multiple real and synthetic datasets demonstrate that our PSD model works well on all three tasks: segmenting object parts, building their hierarchical structure, and capturing their motion distributions.

## 1 Introduction

What makes an object an object? Researchers in cognitive science have made profound investigations into this fundamental problem; results suggest that humans, even young infants, recognize objects as continuous, integrated regions that move together (Carey, 2009; Spelke & Kinzler, 2007). Watching objects move, infants gradually build the internal notion of objects in their mind. The whole process requires little external supervision from experts.

Motion gives us not only the concept of objects and parts, but also their hierarchical structure. The classic study from Johansson (1973) reveals that humans recognize the structure of a human body from a few moving dots representing the keypoints on a human skeleton. This connects to the classic Gestalt theory in psychology (Koffka, 2013), which argues that human perception is holistic and generative, explaining scenes as a whole instead of in isolation. In addition to being unsupervised and hierarchical, our perception gives us concepts that are fully interpretable and disentangled. With an object-based representation, we are able to reason about object motion, predict what is going to happen in the near future, and imagine counterfactuals like "what happens if?" (Spelke & Kinzler, 2007)

How can we build machines of such competency? Would that be possible to have an artificial system that learns an interpretable, hierarchical representation with system dynamics, purely from raw visual data with no human annotations? Recent research in unsupervised and generative deep representation learning has been making progress along this direction: there have been models that efficiently explain multiple objects in a scene (Huang & Murphy, 2015; Eslami et al., 2016), some simultaneously learning an interpretable representation (Chen et al., 2016). Most existing models however either do not produce a structured, hierarchical object representation, or do not characterize system dynamics.

In this paper, we propose a novel formulation that learns an interpretable, hierarchical object representation and scene dynamics by predicting the future. Our model requires no human annotations, learning purely from unlabeled videos of paired frames. During training, the model sees videos of objects moving; during testing, it learns to recognize and segment each object and its parts, build their hierarchical structure, and model their motion distribution for future frame synthesis, all from a single image.

---

* indicates equal contributions. Correspondence to: jiajunwu@mit.edu

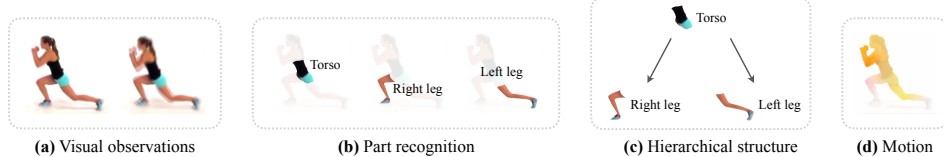

| (a) Visual observations | (b) Part recognition | (c) Hierarchical structure | (d) Motion |

Figure 1: Observing human moving, humans are able to perceive disentangled object parts, understand their hierarchical structure, and capture their corresponding motion fields (without any annotations).

Our model, named Parts, Structure, and Dynamics (PSD), learns to recognize the object parts via a layered image representation. PSD learns their hierarchy via a structural descriptor that composes low-level concepts into a hierarchical structure. Formulated as a fully differentiable module, the structural descriptor can be end-to-end trained within a neural network. PSD learns to model the system dynamics by predicting the future.

We evaluate our model in many possible ways. On real and synthetic datasets, we first examine its ability in learning the concept of objects and segmenting them. We then compute the likelihood that it correctly captures the hierarchical structure in the data. We finally validate how well it characterizes object motion distribution and predicts the future. Our system works well on all these tasks, with minimal input requirement (two frames during training, and one during testing). While previous state-of-the-art methods that jointly discover objects, relations, and predict future frames only work on binary images of shapes and digits, our PSD model works well on complex real-world RGB images and requires fewer input frames.

## 2 RELATED WORK

Our work is closely related to the research on learning an interpretable representation with a neural network (Hinton & Van Camp, 1993; Kulkarni et al., 2015b; Chen et al., 2016; Higgins et al., 2017; 2018). Recent papers explored using deep networks to efficiently explain an object (Kulkarni et al., 2015a; Rezende et al., 2016; Chen et al., 2018), a scene with multiple objects (Ba et al., 2015; Huang & Murphy, 2015; Eslami et al., 2016), or sequential data (Li & Mandt, 2018; Hsu et al., 2017). In particular, Chen et al. (2016) proposed to learn a disentangled representation without direct supervision. Wu et al. (2017) studied video de-animation, building an object-based, structured representation from a video. Higgins et al. (2018) learned an implicit hierarchy of abstract concepts from a few symbol-image pairs. Compared with these approaches, our model not only learns to explain observations, but also build a dynamics model that can be used for future prediction.

There have been also extensive research on hierarchical motion decomposition (Ross & Zemel, 2006; Ross et al., 2010; Grundmann et al., 2010; Xu et al., 2012; Flores-Mangas & Jepson, 2013; Jain et al., 2014; Ochs et al., 2014; Pérez-Rúa et al., 2016; Gershman et al., 2016; Esmaeili et al., 2018). These papers focus on segment objects or parts from videos and infer their hierarchical structure. In this paper, we propose a model that learns to not only segment parts and infer their structure, but also to capture each part's dynamics for synthesizing possible future frames.

Physical scene understanding has attracted increasing attention in recent years (Fragkiadaki et al., 2016; Battaglia et al., 2016; Chang et al., 2017; Finn et al., 2016; Ehrhardt et al., 2017; Shao et al., 2014). Researchers have attempted to go beyond the traditional goals of high-level computer vision, inferring "what is where", to capture the physics needed to predict the immediate future of dynamic scenes, and to infer the actions an agent should take to achieve a goal. Most of these efforts do not attempt to learn physical object representations from raw observations. Some systems emphasize learning from pixels but without an explicitly object-based representation (Fragkiadaki et al., 2016; Agrawal et al., 2016), which makes generalization challenging. Others learn a flexible model of the dynamics of object interactions, but assume a decomposition of the scene into physical objects and their properties rather than learning directly from images (Chang et al., 2017; Battaglia et al., 2016; Kipf et al., 2018). A few very recent papers have proposed to jointly learn a perception module and a dynamics model (Watters et al., 2017; Wu et al., 2017; van Steenkiste et al., 2018). Our model moves further by simultaneously discovering the hierarchical structure of object parts.

Another line of related work is on future state prediction in either image pixels (Xue et al., 2016; Mathieu et al., 2016; Lotter et al., 2017; Lee et al., 2018; Balakrishnan et al., 2018b) or object trajectories (Kitani et al., 2017; Walker et al., 2016). Some of these papers, including our model, draw insights from classical computer vision research on layered motion representations (Wang &

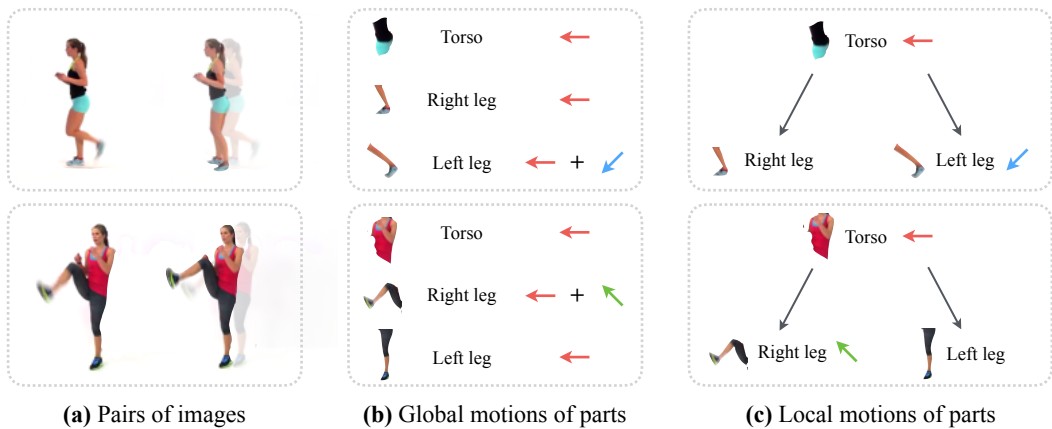

**(a)** Pairs of images      **(b)** Global motions of parts      **(c)** Local motions of parts

Figure 2: Knowing that the legs are part of human body, the legs' motion can be decomposed as the sum of the body's motion and the legs' local motion.

Adelson, 1993). These papers often fail to model the object hierarchy. There has also been abundant research making use of physical models for human or scene tracking (Salzmann & Urtasun, 2011; Kyriazis & Argyros, 2013; Vondrak et al., 2013; Brubaker et al., 2009). Compared with these papers, our model learns to discover the hierarchical structure of object parts purely from visual observations, without resorting to prior knowledge.

## 3 FORMULATION

By observing objects move, we aim to learn the concept of object parts and their relationships. Take human body as an example (Figure 1). We want our model to parse human parts (e.g., torso, hands, and legs) and to learn their structure (e.g., hands and legs are both parts of the human body).

Formally, given a pair of images $\{\mathcal{I}_1, \mathcal{I}_2\}$, let $\mathcal{M}$ be the Lagrangian motion map (i.e. optical flow). Consider a system that learns to segment object parts and to capture their motions, without modeling their structure. Its goal is to find a segment decomposition of $\mathcal{I}_1 = \{\mathcal{O}_1, \mathcal{O}_2, \dots, \mathcal{O}_n\}$, where each segment $\mathcal{O}_k$ corresponds to an object part with distinct motion. Let $\{\mathcal{M}_1^{\mathrm{g}}, \mathcal{M}_2^{\mathrm{g}}, \dots, \mathcal{M}_n^{\mathrm{g}}\}$ be their corresponding motions.

Beyond that, we assume that these object parts form a hierarchical tree structure: each part $k$ has a parent $p_k$, unless itself is the root of a motion tree. Its motion $\mathcal{M}_k^{\mathrm{g}}$ can therefore be decomposed into its parent's motion $\mathcal{M}_{p_k}^{\mathrm{g}}$ and a local motion component $\mathcal{M}_k^{\mathrm{l}}$ within its parent's reference frame. Specifically, $\mathcal{M}_k^{\mathrm{g}} = \mathcal{M}_{p_k}^{\mathrm{g}} + \mathcal{M}_k^{\mathrm{l}}$, if $k$ is not a root. Here we make use of the fact that Lagrangian motion components $\mathcal{M}_k^{\mathrm{l}}$ and $\mathcal{M}_{p_k}^{\mathrm{g}}$ are additive.

Figure 2 gives an intuitive example: knowing that the legs are part of human body, the legs' motion can be written as the sum of the body's motion (e.g., moving to the left) and the legs' local motion (e.g., moving to lower or upper left). Therefore, the objective of our model is, in addition to identifying the object components $\{\mathcal{O}_k\}$, learning the hierarchical tree structure $\{p_k\}$ to effectively and efficiently explain the object's motion.

Such an assumption makes it possible to decompose the complex object motions into simple and disentangled local motion components. Reusing local components along the hierarchical structure helps to reduce the description length of the motion map $\mathcal{M}$. Therefore, such a decomposition should naturally emerge within a design with information bottleneck that encourages compact, disentangled representations. In the next section, we introduce the general philosophy behind our model design and the individual components within.

## 4 METHOD

In this section, we discuss our approach to learn the disentangled, hierarchical representation. Our model learns by predicting future motions and synthesizing future frames without manual annotations. Figure 3 shows an overview of our Parts, Structure, and Dynamics (PSD) model.

### 4.1 OVERVIEW

Motion can be decomposed in a layer-wise manner, separately modeling different object component's movement (Wang & Adelson, 1993). Motivated by this, our model first decomposes the input frame

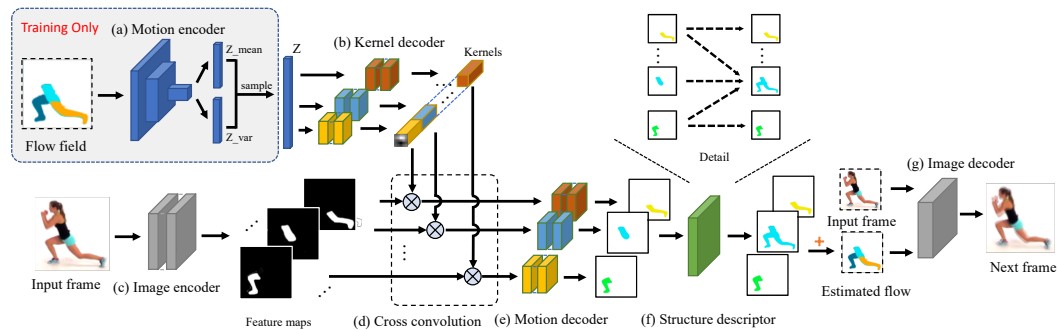

Figure 3: Our PSD model has seven components: **(a)** motion encoder; **(b)** kernel decoder; **(c)** image encoder; **(d)** cross convolution; **(e)** motion decoder; **(f)** structural descriptor; and **(g)** image decoder.

$\mathcal{I}_1$ into multiple feature maps using an *image encoder* (Figure 3c). Intuitively, these feature maps correspond to separate object components. Our model then performs convolutions (Figure 3d) on these feature maps using separate kernels obtained from a *kernel decoder* (Figure 3b), and synthesizes the local motions $\mathcal{M}_k^l$ of separate object components with a *motion decoder* (Figure 3e). After that, our model employs a *structural descriptor* (Figure 3f) to recover the global motions $\mathcal{M}_k^g$ from local motions $\mathcal{M}_k^l$, and then compute the overall motion $\mathcal{M}$. Finally, our model uses an *image decoder* (Figure 3g) to synthesize the next frame $\mathcal{I}_2$ from the input frame $\mathcal{I}_1$ and the overall motion $\mathcal{M}$.

Our PSD model can be seen as a conditional variational autoencoder. During training, it employs an additional *motion encoder* (Figure 3a) to encode the motion into the latent representation $z$; during testing, it instead samples the representation $z$ from its prior distribution $p_z(z)$, which is assumed to be a multivariate Gaussian distribution, where each dimension is *i.i.d.*, zero-mean, and unit-variance. We emphasize the different behaviors of training and testing in Algorithm 1 and 2.

---

**Algorithm 1** Training PSD

**Inputs**: **a pair of** frames $\{\mathcal{I}_1, \hat{\mathcal{I}}_2\}$.
**Outputs**: reconstructed second frame $\mathcal{I}_2$.

$\mathcal{F} = $ image_encoder$(\mathcal{I}_1)$.
$\hat{\mathcal{M}} = $ optical_flow$(\mathcal{I}_1, \hat{\mathcal{I}}_2)$.
$(z_{\text{mean}}, z_{\text{var}}) = $ motion_encoder$(\hat{\mathcal{M}})$.
Randomly sample $z$ from $\mathcal{N}(z_{\text{mean}}, z_{\text{var}})$.

**for** $k = 1$ to $d$ **do**
  $\mathcal{K}_k = $ kernel_decoder$_k(z_k)$.
  $\hat{\mathcal{F}}_k = $ cross_convolution$(\mathcal{F}_k, \mathcal{K}_k)$.
  $\mathcal{M}_k^l = $ motion_decoder$_k(\hat{\mathcal{F}}_k)$.
**end for**

$\{\mathcal{M}_k^g\} = $ structural_descriptor$(\{\mathcal{M}_k^l\}, \mathcal{S})$.
$\mathcal{I}_2 = $ image_decoder$(\mathcal{I}_1, \sum_{k=1}^d \mathcal{M}_k^g)$.

---

**Algorithm 2** Evaluating PSD

**Inputs**: **a single** frame $\mathcal{I}_1$.
**Outputs**: predicted future frame $\mathcal{I}_2$.

$\mathcal{F} = $ image_encoder$(\mathcal{I}_1)$.

Randomly sample $z$ from $\mathcal{N}(0, 1)$.

**for** $k = 1$ to $d$ **do**
  $\mathcal{K}_k = $ kernel_decoder$_k(z_k)$.
  $\hat{\mathcal{F}}_k = $ cross_convolution$(\mathcal{F}_k, \mathcal{K}_k)$.
  $\mathcal{M}_k^l = $ motion_decoder$_k(\hat{\mathcal{F}}_k)$.
**end for**

$\{\mathcal{M}_k^g\} = $ structural_descriptor$(\{\mathcal{M}_k^l\}, \mathcal{S})$.
$\mathcal{I}_2 = $ image_decoder$(\mathcal{I}_1, \sum_{k=1}^d \mathcal{M}_k^g)$.

---

## 4.2 NETWORK STRUCTURE

We now introduce each component.

**Dimensionality.** The hyperparameter $d$ is set to 32, which determines the *maximum* number of objects we are able to deal with. During training, the variational loss encourages our model to use as few dimensions in the latent representation $z$ as possible, and consequently, there will be only a few dimensions learning useful representations, each of which correspond to one particular object, while all the other dimensions will be very close to the Gaussian noise.

**Motion Encoder.** Our motion encoder takes the flow field $\hat{\mathcal{M}}$ between two consecutive frames as input, with resolution of 128×128. It applies seven convolutional layers with number of channels

$\{16, 16, 32, 32, 64, 64, 64\}$, kernel sizes $5\times5$, and stride sizes $2\times2$. Between convolutional layers, there are batch normalizations (Ioffe & Szegedy, 2015), Leaky ReLUs (Maas et al., 2013) with slope 0.2. The output will have a size of $64\times1\times1$. Then it is reshaped into a $d$-dimensional mean vector $z_{\text{mean}}$ and a $d$-dimensional variance vector $z_{\text{var}}$. Finally, the latent motion representation $z$ is sampled from $\mathcal{N}(z_{\text{mean}}, z_{\text{var}})$.

**Kernel Decoder.** Our kernel decoder consists of $d$ separate fully connected networks, decoding the latent motion representation $z$ to the convolutional kernels of size $d\times5\times5$. Therefore, each kernel corresponds to one dimension in the latent motion representation $z$. Within each network, we make uses four fully connected layers with number of hidden units $\{64, 128, 64, 25\}$. In between, there are batch normalizations and ReLU layers.

**Image Encoder.** Our image encoder applies six convolutional layers to the image, with number of channels $\{32, 32, 64, 64, 32, 32\}$, kernel sizes $5\times5$, two of which have strides sizes $2\times2$. The output will be a 64-channel feature map. We then upsample the feature maps by $4\times$ with nearest neighbor sampling, and finally, the resolution of feature maps will be $128\times128$.

**Cross Convolution.** The cross convolution layer (Xue et al., 2016) applies the convolutional kernels learned by the kernel decoder to the feature maps learned by the image encoder. Here, the convolution operations are carried out in a channel-wise manner (also known as depth-wise separable convolutions in Chollet (2017)): it applies each of the $d$ convolutional kernels to its corresponding channel in the feature map. The output will be a $d$-channel transformed feature map.

**Motion Decoder.** Our motion decoder takes the transformed feature map as input and estimates the $x$-axis and $y$-axis motions separately. For each axis, the network applies two $9\times9$, two $5\times5$ and two $1\times1$ depthwise separable convolutional layers, all with 32 channels. We stack the outputs from two branches together. The output motion will have a size of $d\times128\times128\times2$. Note that the local motion $\mathcal{M}_k^{\text{l}}$ is determined by $z_k$ only.

**Structural Descriptor.** Our structural descriptor recovers the global motions $\{\mathcal{M}_k^{\text{g}}\}$ from the local motions $\{\mathcal{M}_k^{\text{l}}\}$ and the hierarchical tree structure $\{p_k\}$ using

$$\mathcal{M}_k^{\text{g}} = \mathcal{M}_k^{\text{l}} + \mathcal{M}_{p_k}^{\text{g}} = \mathcal{M}_k^{\text{l}} + \left(\mathcal{M}_{p_k}^{\text{l}} + \mathcal{M}_{p_{p_k}}^{\text{g}}\right) = \cdots \tag{1}$$

$$= \mathcal{M}_k^{\text{l}} + \sum_{i\neq k} [i \in P_k] \cdot \mathcal{M}_i^{\text{l}}, \quad \text{where } P_k \text{ is the set of ancestors of } \mathcal{O}_k. \tag{2}$$

Then, we define the structural matrix $\mathcal{S}$ as $\mathcal{S}_{ik} = [i \in P_k]$, where each binary indicator $\mathcal{S}_{ik}$ represents whether $\mathcal{O}_i$ is an ancestor of $\mathcal{O}_k$. This is what we aim to learn, and it is shared across different data points. In practice, we relax the binary constraints on $\mathcal{S}$ to $[0, 1]$ to make this module differentiable: $\mathcal{S}_{ik} = \text{sigmoid}(\mathcal{W}_{ik})$, where $\mathcal{W}_{ik}$ are trainable parameters. Finally, the overall motion can be simply computed as $\mathcal{M} = \sum_k \mathcal{M}_k^{\text{g}}$.

**Image Decoder.** Given the input frame $\mathcal{I}_1$ and the predicted overall motion $\mathcal{M}$, we employ the U-Net (Ronneberger et al., 2015) as our image decoder to synthesize the future image frame $\mathcal{I}_2$.

### 4.3 TRAINING DETAILS

Our objective function $\mathcal{L}$ is a weighted sum over three separate components:

$$\mathcal{L} = \mathcal{L}_{\text{recon}} + \beta \cdot \mathcal{L}_{\text{reg}} + \gamma \cdot \mathcal{L}_{\text{struct}}, \quad \text{where } \beta \text{ and } \gamma \text{ are two weighting factors.} \tag{3}$$

The first component is the *pixel-wise reconstruction loss*, which enforces our model to accurately estimate the motion $\mathcal{M}$ and synthesize the future frame $\mathcal{I}_2$. We have $\mathcal{L}_{\text{recon}} = \|\mathcal{M} - \hat{\mathcal{M}}\|_2 + \alpha \cdot \|\mathcal{I}_2 - \hat{\mathcal{I}}_2\|_2$, where $\alpha$ is a weighting factor (which is set to $10^3$ in our experiments).

The second component is the *variational loss*, which encourages our model to use as few dimensions in the latent representation $z$ as possible (Xue et al., 2016; Higgins et al., 2017). We have $\mathcal{L}_{\text{reg}} = \mathcal{D}_{\text{KL}}\big(\mathcal{N}(z_{\text{mean}}, z_{\text{var}}) \,\|\, p_z(z)\big)$, where $\mathcal{D}_{\text{KL}}(\cdot \,\|\, \cdot)$ is the KL-divergence, and $p_z(z)$ is the prior distribution of the latent representation (which is set to normal distribution in our experiments).

The last component is the *structural loss*, which encourages our model to learn the hierarchical tree structure so that it helps the motions $\mathcal{M}^{\text{l}}$ be represented in an efficient way: $\mathcal{L}_{\text{struct}} = \sum_{k=1}^d \|\mathcal{M}_k^{\text{l}}\|_2$. Note that we apply the structural loss on local motion fields, not on the structural matrix. In this way, the structural loss serves as a regularization, encouraging the motion field to have small values.

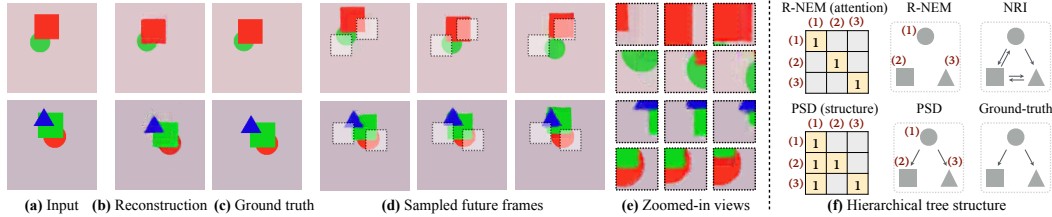

**(a)** Input **(b)** Reconstruction **(c)** Ground truth **(d)** Sampled future frames **(e)** Zoomed-in views **(f)** Hierarchical tree structure

Figure 4: Results of synthesizing future frames **(a-e)** and learning hierarchical structure **(f)** on shapes

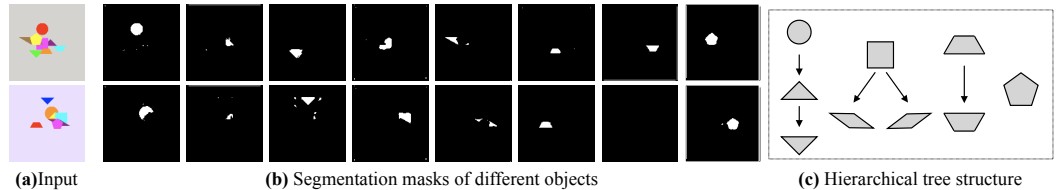

**(a)** Input **(b)** Segmentation masks of different objects **(c)** Hierarchical tree structure

Figure 5: Results of segmenting different shapes **(b)** and learning hierarchical tree structure **(c)** on a dataset with more shapes.

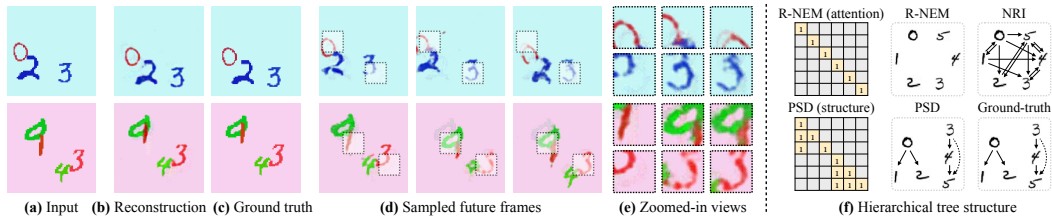

**(a)** Input **(b)** Reconstruction **(c)** Ground truth **(d)** Sampled future frames **(e)** Zoomed-in views **(f)** Hierarchical tree structure

Figure 6: Results of synthesizing future frames **(a-e)** and learning hierarchical structure **(f)** on digits.

We implement our PSD model in PyTorch (Paszke et al., 2017). Optimization is carried out using ADAM (Kingma & Ba, 2015) with $\beta_1 = 0.9$ and $\beta_2 = 0.999$. We use a fixed learning rate of $10^{-3}$ and mini-batch size of 32. We propose the two-stage optimization schema, which first learns the disentangled and then learns the hierarchical representation.

In the first stage, we encourage the model to learn a *disentangled* representation (without structure). We set the $\gamma$ in Equation 3 to 0 and fix the structural matrix $\mathcal{S}$ to the identity $\mathcal{I}$. The $\beta$ in Equation 3 is the same as the one in the $\beta$-VAE (Higgins et al., 2017), and therefore, larger $\beta$'s encourage the model to learn a more disentangled representation. We first initialize the $\beta$ to 0.1 and then adaptively double the value of $\beta$ when the reconstruction loss reaches a preset threshold.

In the second stage, we train the model to learn the *hierarchical* representation. We fix the weights of motion encoder and kernel decoder, and set the $\beta$ to 0. We initialize the structural matrix $\mathcal{S}$, and optimize it with the image encoder and motion decoder jointly. We adaptively tune the value of $\gamma$ in the same way as the $\beta$ in the first stage.

## 5 EXPERIMENTS

We evaluate our model on three diverse settings: i) simple yet nontrivial shapes and digits, ii) Atari games of basketball playing, and iii) real-world human motions.

### 5.1 MOVEMENT OF SHAPES AND DIGITS

We first evaluate our method on shapes and digits. For each dataset, we rendered totally 100,000 pairs for training and 10,000 for testing, with random visual appearance (i.e., sizes, positions, and colors).

For the shapes dataset, we use three types of shapes: circles, triangles and squares. Circles always move diagonally, while the other two shapes' movements consist of two sub-movements: moving together with circles and moving in their own directions (triangles horizontally, and squares vertically). Figure A3 demonstrates the motion distributions of each shape. The complex global motions (after *structure descriptor*) are decomposed into several simple local motions (before *structure descriptor*). These local motions are much easier to represent.

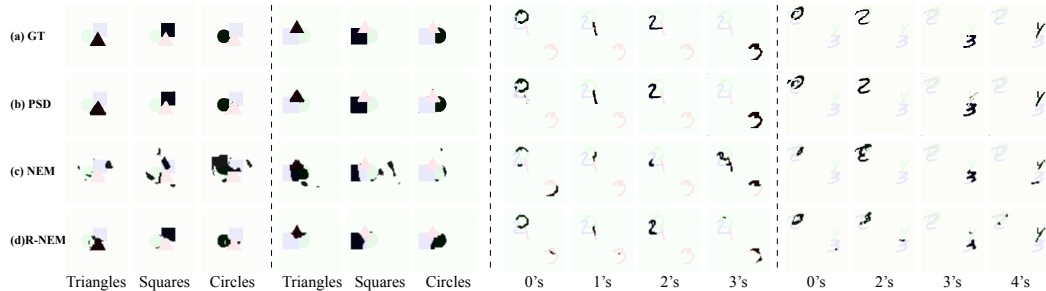

Figure 7: Qualitative results of object segmentation on shapes and digits: **(a)** Ground-truth; **(b)** PSD; **(c)** NEM; and **(d)** R-NEM. In this visualization, we superimpose the segmentation masks on images.

|  | Shapes | | | Digits | | | | | |
|---|---|---|---|---|---|---|---|---|---|
|  | Circles | Squares | Triangles | 0's | 1's | 2's | 3's | 4's | 5's |
| NEM | 0.368 | 0.457 | 0.348 | 0.470 | 0.229 | 0.322 | 0.512 | 0.295 | 0.251 |
| R-NEM | 0.540 | 0.559 | 0.583 | 0.323 | 0.416 | 0.339 | 0.448 | 0.352 | 0.326 |
| PSD (ours) | **0.935** | **0.816** | **0.905** | **0.750** | **0.742** | **0.739** | **0.739** | **0.472** | **0.641** |

Table 1: Quantitative results (IoUs) of object segmentation on shapes and digits.

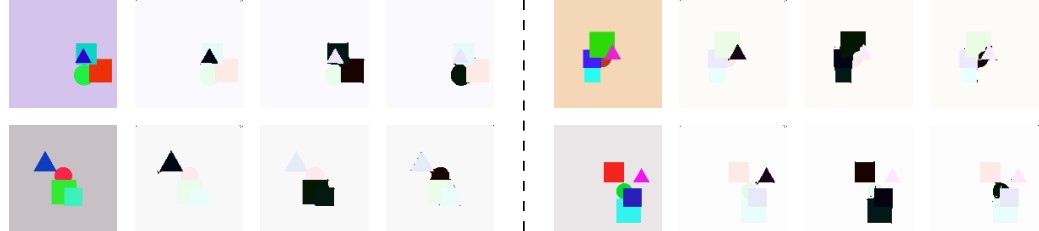

**(a)** Test on original dataset with 2 squares     **(b)** Generalize to new dataset with 3 squares

Figure 8: Qualitative results of object segmentation when generalizing to new dataset.

We also construct an additional dataset with up to nine different shapes. We assign these shapes into four different groups: i) square and two types of parallelograms, ii) circle and two types of triangles, iii) two types of trapezoids, and iv) pentagon. The movements of shapes in the same group have intrinsic relations, while shapes in different groups are independent of each other. These nine shapes have their own different motion direction. In the first group, the tree structure is the same as that of our original shapes dataset: replacing circles with squares, triangles with left parallelograms, and squares with right parallelograms. In the second group, circle and two types of triangles form a chain-like structure, which is similar to the one in our digits dataset. In the third group, the structure is a chain contains two types of trapezoids. In the last group, there is only a pentagon.

As for the digits dataset, we use six types of hand-written digits from MNIST (LeCun et al., 1998). These digits are divided into two groups: 0's, 1's and 2's are in the first group, and 3's, 4's and 5's in the second group. The movements of digits in the same group have some intrinsic relations, while digits in different groups are independent of each other. In the first group, the tree structure is the same as that of our shapes dataset: replacing circles with 0's, triangles with 1's, and squares with 2's. The second group has a chain-like structure: 3's move diagonally, 4's move together with 3' and move horizontally at the same time, and 5's move with 4's and move vertically at the same time.

After training, our model should be able to synthesize future frames, segment different objects (i.e., shapes and digits), and discover the relationship between these objects.

**Future Prediction.** In Figure 4d and Figure 6d, we present some qualitative results of synthesizing future frames. Our PSD model captures the different motion patterns for each object and synthesizes multiple possible future frames. Figure A3 summarizes the distribution of sampled motion of these shapes; our model learns to approximate each shape's dynamics in the training set.

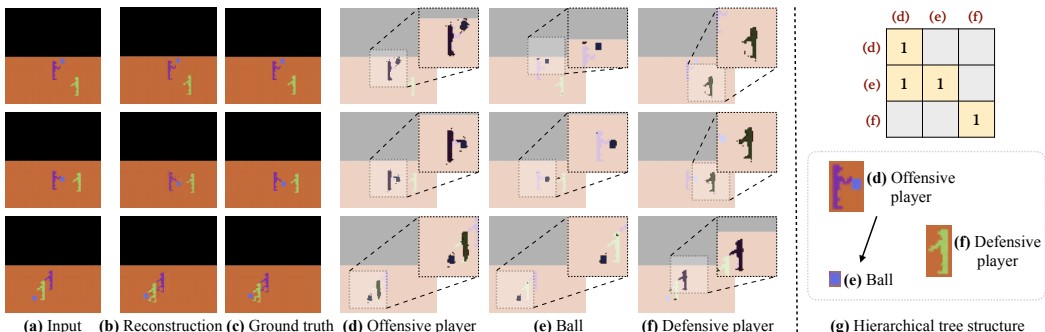

(a) Input    (b) Reconstruction    (c) Ground truth    (d) Offensive player    (e) Ball    (f) Defensive player    (g) Hierarchical tree structure

Figure 9: Results of segmenting objects (**d-f**) and learning hierarchical structure (**g**) on Atari games.

**Latent Representation.** After analyzing the representation $z$, we observe that its intrinsic dimensionality is extremely sparse. On the shapes dataset, there are three dimensions learning meaningful representations, each of which correspond to one particular shape, while all the other dimensions are very close to the Gaussian noise. Similarly, on digits dataset, there are six dimensions, corresponding to different digits. In further discussions, we will only focus on these meaningful dimensions.

**Object Segmentation.** For each meaningful dimension, the feature map can be considered as the segmentation mask of one particular object (by thresholding). We evaluate our model's ability on learning the concept of objects and segmenting them by computing the *intersection over union* (IoU) between model's prediction and the ground-truth instance mask. We compare our model with *Neural Expectation Maximization* (NEM) proposed by Greff et al. (2017) and *Relational Neural Expectation Maximization* (R-NEM) proposed by van Steenkiste et al. (2018). As these two methods both take a sequence of frames as inputs, we feed two input frames repetitively ($\mathcal{I}_1, \mathcal{I}_2, \mathcal{I}_1, \mathcal{I}_2, \mathcal{I}_1, \mathcal{I}_2, ...$) into these models for fair comparison. Besides, as these methods do not learn the correspondence of objects across data points, we manually iterate all possible mappings and report the one with the best performance.

We present qualitative results in Figure 7 and Figure 5b, and quantitative results in Table 1. Our PSD model significantly outperforms two baselines. In particular, R-NEM and our PSD model focus on complementary topics: R-NEM learns to identify instances through temporal reasoning, using signals across the entire video to group pixels into objects; our PSD model learns the appearance prior of objects: by watching how they move, it learns to recognize how object parts can be grouped based on their appearance and can be applied on static images. As the videos in our dataset has only two frames, temporal signals alone are often not enough to tell objects apart. This explains the less compelling results from R-NEM. We included a more systematic study in Section A.3 to verify that.

To evaluate the generalization ability, we train our PSD model on a dataset with two squares, among other shapes, and test it on a dataset with three squares. In each piece of data, all squares move together and have the same motion. Other settings are the same as the original shapes dataset. Figure 8 shows segmentation results on these two datasets. Our model generalizes to recognize the three squares simultaneously, despite having seen up to two in training.

**Hierarchical Structure.** To discover the tree structure between these dimensions, we binarize the structural matrix $S_{ik}$ by a threshold of 0.5 and recover the hierarchical structure from it. We compare our PSD model with R-NEM and *Neural Relational Inference* (NRI) proposed by Kipf et al. (2018). As the NRI model requires objects' feature vectors (i.e., location and velocity) as input, we directly feed the coordinates of different objects in and ask it to infer the underlying interaction graph. In Figure 4f and Figure 6f, we visualize the hierarchical tree structure obtained from these models. Our model is capable of discovering the underlying structure; while two baselines fail to learn any meaningful relationships. This might be because NRI and R-NEM both assume that the system dynamics is fully characterized by their current states and interactions, and therefore, they are not able to model the uncertainties in the system dynamics. On the challenging dataset with more shapes, our PSD model is still able to discover the underlying structure among them (see Figure 5c).

## 5.2 ATARI GAMES OF PLAYING BASKETBALL

We then evaluate our model on a dataset of Atari games. In particular, we select the Basketball game from the Atari 2600. In this game, there are two players competing with each other. Each player can move in eight different directions. The *offensive* player constantly dribbles the ball and throws the

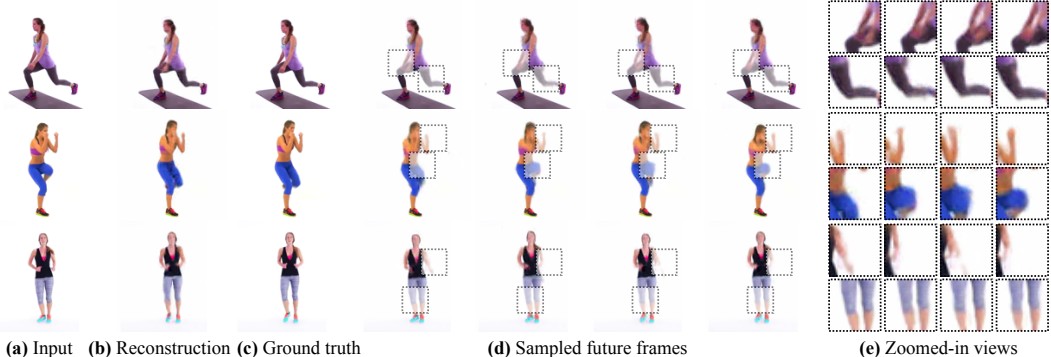

**(a)** Input    **(b)** Reconstruction    **(c)** Ground truth      **(d)** Sampled future frames      **(e)** Zoomed-in views

Figure 10: Qualitative results of synthesizing future frames on real-world human motions (exercise).

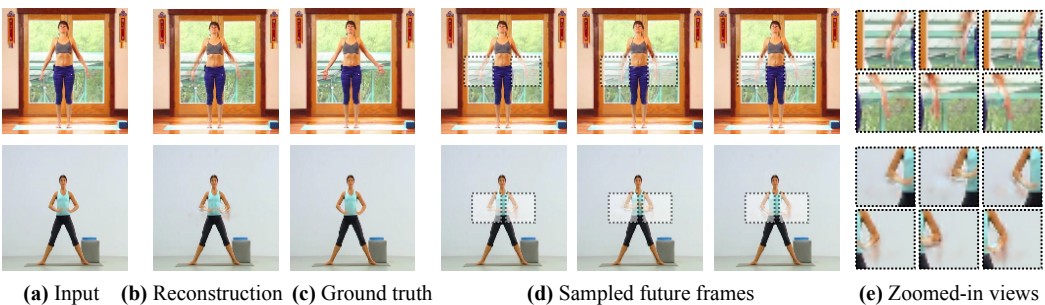

**(a)** Input    **(b)** Reconstruction    **(c)** Ground truth      **(d)** Sampled future frames      **(e)** Zoomed-in views

Figure 11: Qualitative results of synthesizing future frames on real-world human motions (yoga).

ball at some moment; while the *defensive* player tries to steal the ball from his opponent player. We download a video of playing this game from YouTube and construct a dataset with 5,000 pairs for training and 500 for testing.

Our PSD model discovers three meaningful dimensions in the latent representation $z$. We visualize the feature maps in these three dimensions in Figure 9. We observe that one dimension (in Figure 9d) is learning the *offensive player with ball*, another (in Figure 9e) is learning the *ball*, and the other (in Figure 9f) is learning the *defensive player*. We construct the hierarchical tree structure among these three dimensions from the structural matrix $\mathcal{S}$. As illustrated in Figure 9g, our PSD model is able to discover the relationship between the ball and the players: the offensive player *controls* the ball. This is because our model observes that the ball always moves along with the offensive player.

### 5.3 MOVEMENT OF HUMANS

We finally evaluate our method on two datasets of real-world human motions: the human exercise dataset used in Xue et al. (2016) and the yoga dataset used in Balakrishnan et al. (2018a). We estimate the optical flows between frames by an off-the-shelf package (Liu, 2009). Compared with previous datasets, these two require much more complicated visual perception, and they have challenging hierarchical structures. In the human exercise dataset, there are 50,000 pairs of frames used for training and 500 for testing. As for the yoga dataset, there are 4,720 pairs of frames for training and 526 for testing.

**Future Prediction.** In Figure 10 and Figure 11, we present qualitative results of synthesizing future frames. Our model is capable of predicting multiple future frames, each with a different motion. We compare with 3DcVAE (Li et al., 2018), which takes one frame as input and predicts the next 16 frames. As our training dataset only has paired frames, for fair comparison, we use the repetition of two frames as input: $(\mathcal{I}_1, \mathcal{I}_2, \mathcal{I}_1, \mathcal{I}_2, ..., \mathcal{I}_1, \mathcal{I}_2)$. We also use the same optical flow (Liu, 2009) for both methods. In Figure 12, the future frames predicted by 3DcVAE have much more artifacts, compared with our PSD model.

**Object Segmentation.** In Figure 13 and Figure 14, we visualize the feature maps corresponding to the active latent dimensions. It turns out that each of these dimensions corresponds to one particular human part: full torsos (13c, 14c), upper torsos (13d), arms (13e), left arms (14d), right arms (14e), right legs (13f, 14g), and left legs (13g, 14f). Note that it is extremely challenging to distinguish

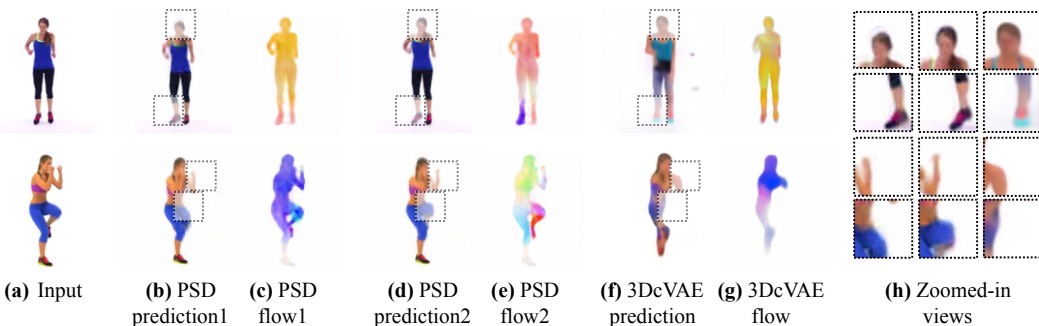

| **(a)** Input | **(b)** PSD prediction1 | **(c)** PSD flow1 | **(d)** PSD prediction2 | **(e)** PSD flow2 | **(f)** 3DcVAE prediction | **(g)** 3DcVAE flow | **(h)** Zoomed-in views |

Figure 12: Comparison of synthesizing future frames between our PSD model and 3DcVAE

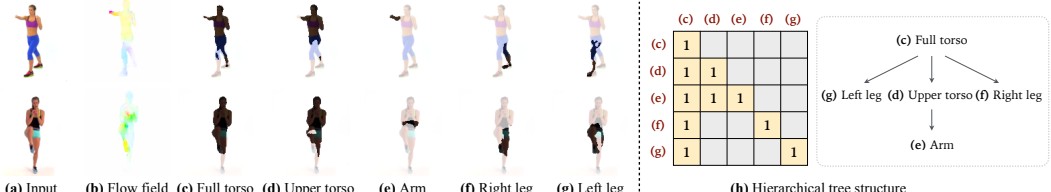

**(a)** Input   **(b)** Flow field   **(c)** Full torso   **(d)** Upper torso   **(e)** Arm   **(f)** Right leg   **(g)** Left leg   **(h)** Hierarchical tree structure

Figure 13: Results of segmenting parts **(c-g)** and learning hierarchical structure **(h)** on human motions.

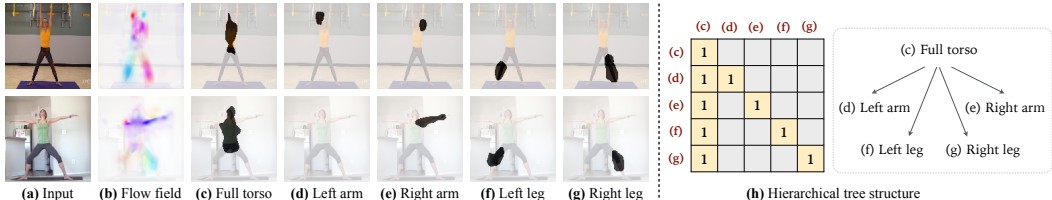

**(a)** Input   **(b)** Flow field   **(c)** Full torso   **(d)** Left arm   **(e)** Right arm   **(f)** Left leg   **(g)** Right leg   **(h)** Hierarchical tree structure

Figure 14: Results of segmenting parts **(c-g)** and learning hierarchical structure **(h)** on human motions.

| | Full torso | Upper torso | Arm | Left leg | Right leg | Overall |
|---|---|---|---|---|---|---|
| NEM | 0.298 | 0.347 | 0.125 | 0.264 | 0.222 | 0.251 |
| R-NEM | 0.321 | 0.319 | 0.220 | 0.294 | 0.228 | 0.276 |
| PSD (ours) | **0.697** | **0.574** | **0.391** | **0.374** | **0.336** | **0.474** |

Table 2: Quantitative results (IoUs) of object segmentation on human exercise dataset.

different parts from motions, because different parts (e.g., arms and legs) might have similar motions (see Figure 13b). R-NEM is not able to segment any meaningful parts, let alone structure, while our PSD model gives imperfect yet reasonable part segmentation results. For quantitative evaluation, we collect the ground truth part segmentation for 30 images and compute the *intersection over union* (IoU) between the ground-truth and the prediction of our model and the other two baselines (NEM, R-NEM). The quantitative results are presented in Table 2. Our PSD model significantly outperforms the two baselines.

**Hierarchical Structure.**   We recover the hierarchical tree structure among these dimensions from the structural matrix $\mathcal{S}$. From Figure 13h, our PSD model is able to discover that the upper torso and the legs are part of the full torso, and the arm is part of the upper torso, and from Figure 14h, our PSD model discovers that the arms and legs are parts the full torso.

## 6   CONCLUSION

We have presented a novel formulation that simultaneously discovers object parts, their hierarchical structure, and the system dynamics from unlabeled videos. Our model uses a layered image representation to discover basic concepts and a structural descriptor to compose them. Experiments suggest that it works well on both real and synthetic datasets for part segmentation, hierarchical structure recovery, and motion prediction. We hope our work will inspire future research along the direction of learning structural object representations from raw sensory inputs.

**Acknowledgements.** We thank Michael Chang and Sjoerd van Steenkiste for helpful discussions and suggestions. This work was supported in part by NSF #1231216, NSF #1447476, ONR MURI N00014-16-1-2007, and Facebook.

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

## A.1 MORE QUALITATIVE RESULTS

In Figure A1, we demonstrate more future prediction results on shapes and digits dataset. In Figure A2, we present several segmentation results on shapes (with more objects), exercise and yoga dataset.

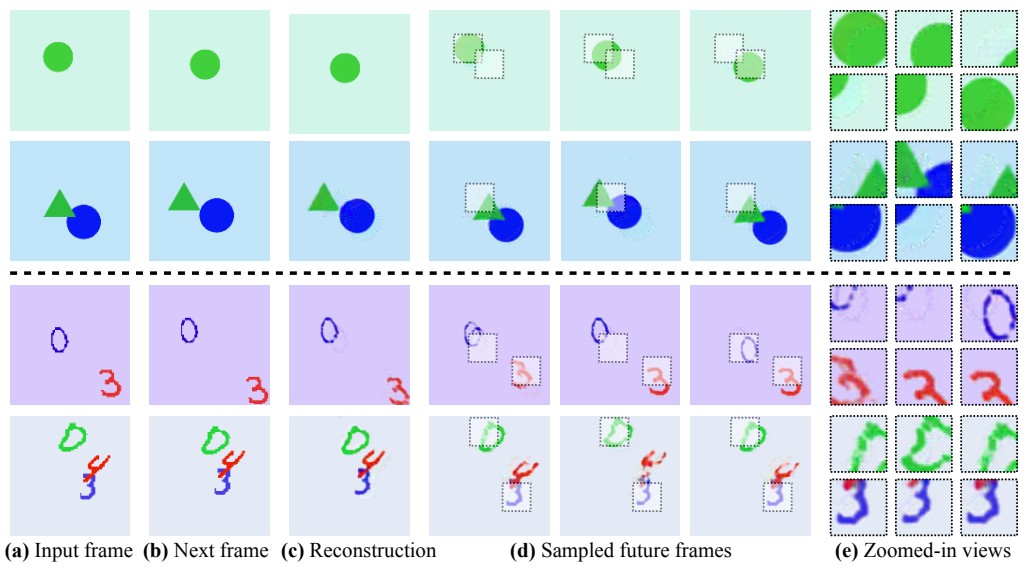

(a) Input frame (b) Next frame (c) Reconstruction    (d) Sampled future frames    (e) Zoomed-in views

Figure A1: Additional results of future prediction on shapes and digits dataset.

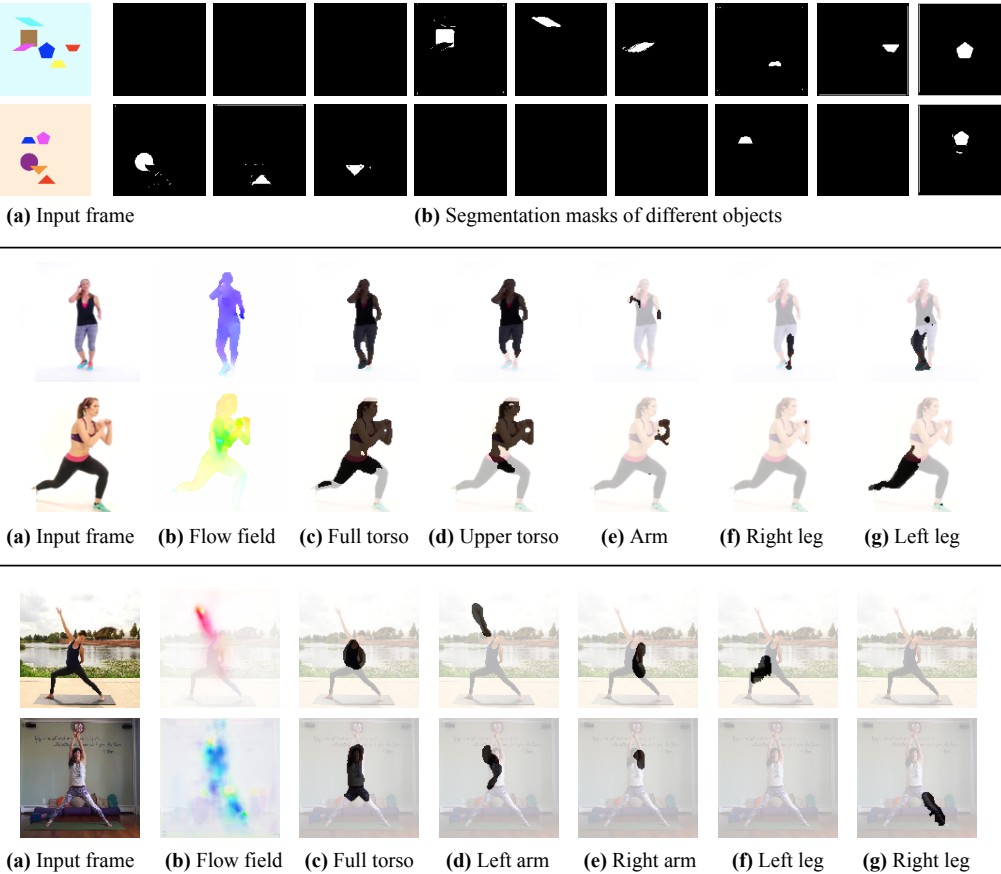

(a) Input frame                        (b) Segmentation masks of different objects

(a) Input frame (b) Flow field (c) Full torso (d) Upper torso    (e) Arm    (f) Right leg    (g) Left leg

(a) Input frame (b) Flow field (c) Full torso (d) Left arm    (e) Right arm    (f) Left leg    (g) Right leg

Figure A2: Additional results of segmentation on shapes, exercise and yoga dataset.

## A.2 MOTION DISTRIBUTION OF SHAPE DATASET

In Figure A3, we demonstrate the motion distributions of each shape.

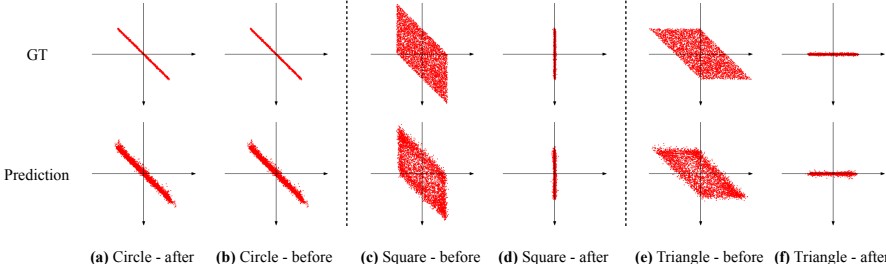

(a) Circle - after    (b) Circle - before    (c) Square - before    (d) Square - after    (e) Triangle - before    (f) Triangle - after

Figure A3: Motion distributions of different shapes before and after the *structure descriptor*. The first row is the ground truth and the second row is the prediction of our model.

## A.3 ADDITIONAL RESULTS OF R-NEM

As mentioned in the main paper, R-NEM and our PSD model focus on complementary topics: R-NEM learns to identify instances through temporal reasoning, using signals across the entire video to group pixels into objects; our PSD model learns the appearance prior of objects: by watching how they move, it learns to recognize how object parts can be grouped based on their appearance and can be applied on static images. As the videos in our dataset has only two frames, temporal signals alone are often not enough to tell objects apart. This may explain the less compelling results from R-NEM.

Here, we include a more systematic study to verify that. We train the R-NEM with three types of inputs: 1) only one frame; 2) two input frames appear repetitively (the setup we used on our dataset, where videos only have two frames); 3) longer videos with 20 sequential frames. Figure A4 and Table A1 show that results on 20-frame input are significantly better than the previous two. R-NEM handles occluded objects with long trajectories, where each object appears without occlusion in at least one of the frames.

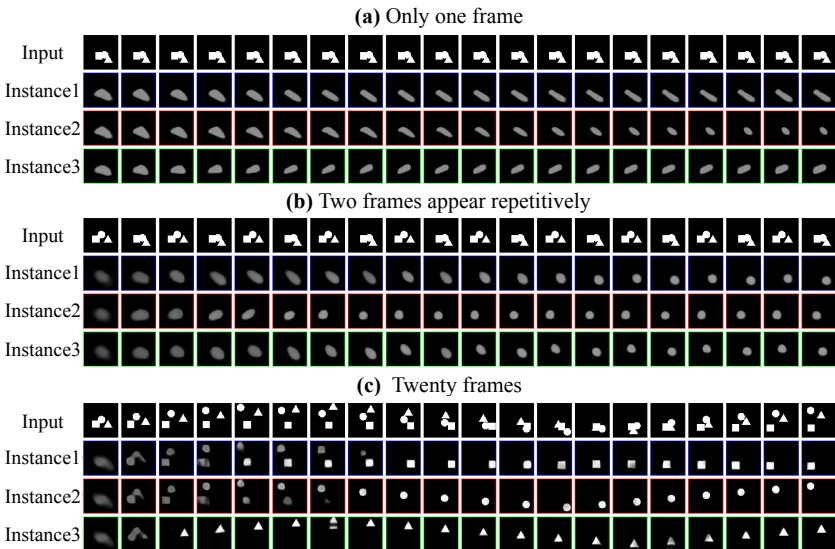

Figure A4: Results of R-NEM on different kinds of datasets.

|  | Circles | Squares | Triangles | Overall |
|---|---|---|---|---|
| 1 frame | 0.418 | 0.511 | 0.559 | 0.501 |
| 2 frames | 0.513 | 0.552 | 0.612 | 0.558 |
| 20 frames | 0.760 | 0.850 | 0.871 | 0.833 |

Table A1: Quantitative results (IoUs) of object segmentation with different types of inputs.

