# OpenReview forum: "Unsupervised Discovery of Parts, Structure, and Dynamics"
_ICLR.cc/2019/Conference_

### Official Review · AnonReviewer3 · 2018-11-02
**Overall interesting approach and well written paper but limited experimental results**

**Rating:** 5
**Confidence:** 3

**Review:**

This paper presents a method for learning about the parts and motion dynamics of a video by trying to predict future frames.  Specifically, a model based on optical flow is defined, noting that the motion of hierarchically related parts are additive.  Flow fields are represented using an encoder/decoder architecture and a binary structural matrix encodes the representations between parts.  This matrix is predicted given the previous frame and flow field.  This is then used to estimate a new flow field and generate a possible future frame.  The system is trained to predict future frames using an L2 loss on the predicted image and motion field and regularized to prefer more compact and parsimonious representations.

The method is applied to synthetic datasets generated by moving shapes or MNIST digits and shown to work well compared to some recent baseline methods for part segmentation and hierarchy representation.  It is also applied and qualitatively evaluated for future frame prediction on an atari video game and human motion sequences.  The qualitative evaluation shows that part prediction is plausible but the results for future frame prediction are somewhat unclear as there are no baseline comparisons for this aspect of the task.

Overall the approach seems very interesting and well motivated.   However, the experimental comparisons are limited and baselines are lacking.  Further, some relevant related work is missing.

Specific concerns:
- Motion segmentation has been studied for a long time in computer vision, a comparison against some of these methods may be warranted.  See, e.g., Mangas-Flores and Jepson, CVPR 2013.
- There is some missing related work on learning part relations.  See, e.g., Ross, et al IJCV 2010 and Ross and Zemel JMLR 2006.
- There is also some missing work on future frame prediction.  In particular, PredNet seems relevant to discuss in the context of this work and as a baseline comparison method.  See Lotter et al ICLR 2017.
- A reasonable baseline might be simply to apply the previous frames motion field to generate the next frame.  This would be a good comparison to include.
- The "Human Studies" section is very unclear.  How is "same tree structure" defined exactly and how were humans asked to annotate the tree structure?  If it's about the hierarchical relationship, then I would expect humans to always be pretty consistent with the hierarchy of body parts and suggests that the model is doing relatively poorly.  If it's some other way, then this needs to be clarified.  Further, how was this study performed?  If this section can't be thoroughly explained it should be removed from the paper as it is at best confusing and potentially very misleading.
- The system only considers a single frame and flow-field for part prediction.  From this perspective, the effectiveness of the method seems somewhat surprising.
- The system takes as input both a frame and a flow field.  I assume that flow field is computed between I0 and I1 and not I1 and I2, however this is never specified anywhere I can find in the manuscript.  If this is not the case, then the problem setup is (almost) trivial.

---

> ### Author Response · Authors · 2018-11-16
> **Our Response to Reviewer 3**
>
> Thank you very much for the constructive comments.
>
> 1. Problem setup and baselines
> We would like to clarify that while our model sees two frames during training, it only takes only a **single** image as input during testing. It segments object parts in the image, infers their structure, and synthesizes multiple future frames based on the inferred segments, structure, and the sampled motion vector z. The task is highly non-trivial, as it requires the model to associate object appearance with their possible motion purely from unannotated data.
>
> Our goal is to learn the prior that ties part structure and dynamics to their appearance. Only with the learned prior, our model can segment object parts and synthesize motion from a single image. For example, the “torso” is always the parent of the “leg”, and the motion of the leg is always affected by the motion of the torso. Therefore, in our datasets, parts always have the same hierarchical tree structure.
>
> We agree with the reviewers that it’d be important to add more baselines on both motion segmentation and future prediction. Note that our model takes a single image for segmentation and future prediction, while most baselines require multi-frame input (e.g., requiring the previous motion field). We will make a comparison with 3DcVAE [1], which only needs one frame as input.
>
> 2. Related work
> Thank you for pointing out the missing related work, which we will cite and discuss.
>
> 3. Human studies
> In our human study, we first explained the definition of hierarchical structure to subjects. We then showed the segmentation masks of our model (Figure 11c-11g) and asked the subjects to generate the tree hierarchy based on the segments.
>
> The ‘same tree structure’ means that corresponding nodes (segments) have the same parent across the two trees. The alternative response provided by the subjects is a two-level hierarchy, putting the arm directly below the full torso instead of the upper torso (Figure 11h).
>
> We agree that the experiment is not well explained and will remove it in the revision.
>
> We have also listed all other planned changes in our general response above. Please don’t hesitate to let us know for any additional comments on the paper or on the planned changes.
>
> Reference:
> [1] Li, Yijun and Fang, Chen and Yang, Jimei and Wang, Zhaowen and Lu, Xin and Yang, Ming-Hsuan. Flow-Grounded Spatial-Temporal Video Prediction from Still Images. In ECCV, 2018.

---

> > ### Author Response · Authors · 2018-12-04
> > **Our Response and Revision**
> >
> > Dear Reviewer 3,
> >
> > Thanks again for your constructive reviews, which have helped us improved the quality and clarity of the paper. In particular, in the revision, we have cited and discussed the suggested related work, included an algorithm box, revised the method section, and updated the figure to better explain the algorithm and its setup (taking two images during training, and only one during testing). Per your suggestion, we have also compared our model with the state of the art (3DcVAE) on future prediction.
> >
> > As the discussion period is about to end, please don’t hesitate to let us know if there are any additional clarifications that we can offer, as we would love to convince you of the merits of the paper. We appreciate your suggestions. Thanks!

---

### Official Review · AnonReviewer1 · 2018-11-03
**Interesting idea**

**Rating:** 7
**Confidence:** 4

**Review:**

The paper describes a method, which learns the hierarchical decomposition of moving objects into parts without supervision, based on prediction of the future. A deep neural network is structured into a sequence of encoders and decoders: the input image is decomposed into objects by a trained head, then motion is estimated from predicted convolutional kernels whose model is trained on optical flow; the latent motion output is encoded into separated motion fields for each object and then composed into a global model with a trainable structured matrix which encodes the part hierarchy. The latent space is stochastic similar to VAEs and trained with similar losses.

Strengths:

The idea is interesting and nicely executed. I particularly appreciated the predicted kernels, and the trainable structure matrix. Although the field of hierarchical motion segmentation is well studied, up to my knowledge this method seems to be the first of its kind based on a fully end-to-end trainable method where the motion estimators, the decomposition and the motion decoders are learned jointly.

The method is evaluated on different datasets including fully synthetic ones with synthetic shapes or based on MNIST; very simple moving humans taken from ATARI games, and realistic humans from two different pose datasets. The motion decomposition is certainly not as good as the definition and the output of a state of the art human pose detector; however, given that the decomposition is discovered, the structure looks pretty good.

Weaknesses

I have two issues with the paper. First of all, although the related work section is rich, the methods based on hierarchical motion decompositions are rarer, although the field is quite large. Below are a couple of references:

Mihir Jain, Jan Van Gemert, Hervé Jégou, Patrick Bouthemy, and Cees GM Snoek. Action localization with tubelets from motion. CVPR, 2014.

Chenliang Xu and Jason J Corso. Evaluation of super-voxel methods for early video processing. CVPR, 2012.

Jue Wang, Bo Thiesson, Yingqing Xu, and Michael Cohen. Image and video segmentation by anisotropic kernel mean shift. ECCV, 2004

Chenliang Xu, Caiming Xiong, and Jason J Corso. Streaming hierarchical video segmentation. ECCV 2012.

Matthias Grundmann, Vivek Kwatra, Mei Han, and Irfan Essa. Efficient hierarchical graph-based video segmentation. CVPR, 2010.

Peter Ochs, Jitendra Malik, and Thomas Brox. Segmentation of moving objects by long term video analysis. IEEE PAMI, 2014.

Discovering motion hierarchies via tree-structured coding of trajectories
Juan-Manuel Pérez-Rúa, Tomas Crivelli, Patrick Pérez, Patrick Bouthemy, BMVC 2016.

Samuel J Gershman, Joshua B Tenenbaum, and Frank Jäkel. Discovering hierarchical motion structure. Vision Research, 2015.

Secondly, the presentation is not perfect. The paper is densely written with lots of information thrown rapidly at the reader. Readers familiar with similar work can understand the paper (I needed a second pass). But many parts could be better formulated and presented.

I understood the equations, but I needed to ignore certain thinks in order to understand them. One of them is the superscript in the motion matrices M, which does not make sense to me. “g” seems to indicate “global” and “l” local, but then again a local parent matrix gets index “g”, and this index seems to switch whether the same node is seen as the current node or the parent of its child.

Figure 3 is useful, but it is hard to make the connection with the notation. Symbols like z, M etc. should be included in the figure.

The three lines after equations 2 and 3 should be rewritten. They are understandable but clumsy. Also, we can guess where the binary constraints come from, but they should be introduced nevertheless.

In essence, the paper is understandable with more efforts than there should be necessary.


Other remarks:

The loss L_struct is L_2, I don’t see how it can favor sparsity. This should be checked and discussed.

A symbolic representation is mentioned in the introduction section. I am not sure that this notion is completely undisputed in science, it should at least not be presented as a fact.

The ATARI dataset seems to be smallish (a single video and 5000 frames only).

---

> ### Author Response · Authors · 2018-11-16
> **Our Response to Reviewer 1**
>
> Thank you very much for the constructive comments.
>
> 1. Presentation
> In the revision, we’ll include a separate paragraph in the related work to discuss hierarchical motion decomposition methods. We’ll revise the method section for a better presentation of the model and the equations. We’ll also revise sentence about symbolic representation and Figure 3 as suggested.
>
> 2. Structural loss
> We apply the structural loss on local motion fields, not on the structural matrix. In this way, the structural loss serves as a regularization, encouraging the motion field to have small values. This is different from the traditional L1 sparseness loss, which encourages values to be 0. Following your suggestion, we’ve also experimented with the L1 loss on the shapes dataset, and found that using L1 or L2 structural loss leads to similar results. We’ll include this discussion into the revision.
>
> 3. Atari dataset
> The purpose of the Atari dataset is to demonstrate that our model works well on a different domain and learns to discover interesting structure (the ball belongs to the offensive player). There, as the concepts and structure are relatively simple, we found that 5000 frames are sufficient for our purpose.
>
> We have also listed all other planned changes in our general response above. Please don’t hesitate to let us know for any additional comments on the paper or on the planned changes.

---

> > ### Author Response · Authors · 2018-12-04
> > **Our Response and Revision**
> >
> > Dear Reviewer 1,
> >
> > Thank you for your constructive review! Based on your suggestion, we have cited and discussed the suggested related work. We also included an algorithm box, revised the method section, and updated the figure to better explain the algorithm and its setup. We hope the revision is now better.
> >
> > As the discussion period is about to end, please don’t hesitate to let us know if there are any additional clarifications that we can offer. We appreciate your suggestions. Thanks!

---

### Official Review · AnonReviewer2 · 2018-11-08
**Interesting and novel works but only tested on simple dataset**

**Rating:** 6
**Confidence:** 3

**Review:**

The paper proposes an unsupervised learning model that learns to (1) disentangle object into parts, (2) predict hierarchical structure for the parts and (3), based on the disentangled parts and the hierarchy, predict motion. The model is trained to predict future frames and motion with L2 loss given current frame and observed motion. The overall objective is similar to a standard VAE.

One interesting module proposed in this work is the structural descriptor which assumes motions are additive and global motion of an object part can be recovered by adding the local motion of this object with the global motions of its parents. The equation can be applied recursively and it generalizes to any arbitrary hierarchy depth.

Pros:
The overall methodology is quite novel and results look good. Merging hierarchical inference into the auto-encoder kind of structure for unsupervised learning is new.
The results are tested on both synthetic and real videos.

Cons:
The idea is only tested on relatively simple dataset. For the synthetic data, the objects only have very restrictive motions (ex. Circles always move diagonally). It is also unclear to me whether all the samples in the dataset share the same hierarchical tree structure or not (For human, it seems like every sample uses the same tree).  If this is the case, then it means you need 100K data to learn one hierarchical relationship for very simple video.
From the human dataset results, since the appearance and motions become so different across videos, making the video clean and making the objects aligned (so that motions are aligned) seems important to make the model work. For example, from figure 11(f)(g), the right and left legs are exchanged for the person on the top. This brings up the concern that the model is fragile to more complicated scenes such that objects are not super aligned and the appearances differ a lot. (ex. Videos of people playing different sports shooting from different views)
Should add more baselines. There are a lot of unsupervised video prediction works which also unsupervisedly disentangle motions and contents.

Others:
The sparsity constraint seems incorrect

---

> ### Author Response · Authors · 2018-11-16
> **Our Response to Reviewer 2**
>
> Thank you very much for the constructive comments.
>
> 1. Baselines
> As you summarized, the goal of this paper is beyond from video prediction: From pairs of unlabeled frames, our model learns to solve three tasks at the same time: 1) learning to segment object parts; 2) learning their hierarchical structure; 3) learning their dynamics for future prediction. During testing, from a single image, our model segments its parts and synthesizes possible future frames.
>
> Most video prediction methods do not learn part hierarchy. The only previous method that attempts to solve the three problems at the same time is RNEM, which we’ve compared with in the paper. During testing, with 20 frames as input, RNEM performs well on binary images (black/white), but does not learn meaningful concepts on grayscale or color images. In comparison, our model performs well on real data, even with cluttered background, using just a single image as input during testing. Our datasets are highly challenging, and our model achieves significant performance gain.
>
>
> Our goal is to learn the prior that ties part structure and dynamics to their appearance. Only with the learned prior, our model can segment object parts and synthesize motion from a single image. For example, the “torso” is always the parent of the “leg”, and the motion of the leg is always affected by the motion of the torso. Therefore, in our datasets, parts always have the same hierarchical tree structure.
>
> While future prediction is not our focus, we agree with the reviewers that it’d be important to add more baselines. Note that our model takes a single image for future prediction, while most video prediction algorithms require multi-frame input. We will make a comparison with 3DcVAE, which only needs one frame as input.
>
> 2. Data-efficiency and robustness
> Our model is data-efficient. When the motion is simple (Atari games), our models learns from only 5K pairs of frames (i.e., 10K images). On real data, our model learns from 9K pairs of images with cluttered backgrounds (the yoga dataset). Our model is robust to unaligned objects: on the shapes and digits datasets, the object positions are random. We agree with the reviewer that discovering hierarchical parts and their motions from purely unlabeled, in-the-wild videos would be an ultimate goal. At the same time, we also believe our model has been making solid and significant progress compared with the state-of-the-art, which, as mentioned above, only works on binary images.
>
> Thanks for the observation on the flipped left/right legs. They are indistinguishable in our current setup---we’re learning purely from motion signals, and these parts have identical motion no matter whether they’re flipped on not. This suggests an important future research direction---how we can develop a model that learns to discover semantically rich concepts from videos, with minimal supervision.
>
> 3. Structural loss
> We apply the structural loss on local motion fields, not on the structural matrix. In this way, the structural loss serves as a regularization, encouraging the motion field to have small values. This is different from the traditional L1 sparseness loss, which encourages values to be 0. We’ve also experimented with the L1 loss on the shapes dataset, and found that using L1 or L2 structural loss leads to similar results. We’ll include this discussion into the revision.
>
> We have also listed all other planned changes in our general response above. Please don’t hesitate to let us know for any additional comments on the paper or on the planned changes.

---

> > ### Author Response · Authors · 2018-12-04
> > **Our Response and Revision**
> >
> > Dear Reviewer 2,
> >
> > We'd like to thank you again for your constructive reviews, which have helped us make the paper better. Based on your review, in the revision, we have revised the method section and discussed more details of the structural loss. We have also included a systematic comparison with the state of the art on both structure recovery (R-NEM) and future prediction (3DcVAE).
> >
> > As the discussion period is about to end, please don’t hesitate to let us know if there are any additional clarifications that we can offer, as we would love to convince you of the merits of the paper. We appreciate your suggestions. Thanks!

---

### Official Review · AnonReviewer4 · 2018-11-08

**Rating:** 6
**Confidence:** 3

**Review:**

==== Review Summary ====

The paper demonstrates an interesting and potentially useful idea.  But much of it is poorely explained, and experimental results are not strongly convincing.  The only numerical evaluations are on a simple dataset that the authors made themselves.  The most interesting claim - that this network can learn unsupervised hierarchical object segmentation based on unlabelled video data -  is not well supported by the paper.

==== Paper Summary ====

This paper presents a deep neural network which learns object Segmentation, Structure, and Dynamics from unlabelled video.  The idea is quite useful, as it is a step towards learning models that can "discover" the concept of objects in a visual scene without any supervision.

The central contributions of this paper are:
(1) To show how one can use coherent motion (the fact that different parts of an object move together) to learn unsupervised object segmentation.
(2) To show how once can learn the relation between objects (e.g. "arm" is part of "body") by observing relative motion between segments.

==== General Feedback ====

The paper would benefit a lot from better explanations and being better tied together (see "Details" below for examples).   Figure captions would benefit from much better explanations and integration with the text - each figure caption should at least describe what the figure is intended to demonstrate.  Variables such as ($\mathcal M$, $\mathcal S$, $\mathcal I$, $\mathcal L$) should be indicated in figures .

Many things would benefit from being defined precisely with Equations.  For example I have no idea how the "soft" structural descriptor S is computed.  Is it (A) a parameter that is shared across data points and learned?  or (B) is it computed per-frame from the network?  And after it is calculated, how are the S_{ik} values (which fall between 0 and 1) used to combine the motion maps?

==== Scientific Questions ===

I'm confused as to what the latent variables z "mean".  It seems strange that there is a 1-d latent variable representing the motion of each part.  Translation of a segment within an image is 2D.  3D if you include planar rotation, then there's scaling motion and out-of-plane rotations, so it seems an odd design choice that motion should be squeezed into a 1D representation.

I find it difficult to assess the quality of the "next frame predictions".  There's lots other literature on next-frame prediction to compare against (e.g. https://arxiv.org/abs/1605.08104).  At least you could compare to a naive baseline of simply shifting pixels based on the optical flow.

I'm confused about how you are learning the "estimated flow".  My impression is that the input flow is calculated between the last 2 frames $\hat M = flow(I_{t-1}, I_t)$.  And that the "estimated" flow is an estimate of $flow(I_{t}, I_{t+1})$.  But in Section 4.3 you seem to indicate that the "estimated" flow is just trained to "reconstruct" the input flow.... In that case why not just feed the input flow directly into the Image Decoder?  What I guess you're doing is trying to Predict the next flow ($flow(I_{t}, I_{t+1})$) but if you're doing this neither Figure 3 nor Section 4.2 indicates this, hence my confusion.

==== Details ====

Figure 3:
----
The "shapes" example is odd, because it's not obvious that there's a structural hierarchy linking the shapes.  Maybe a "torso/left arm/right arm" would be better for illustrative purposes?
It would be helpful to put the variable names ($\mathcal M_k$, etc) on the figure.
Should add a second arrow coming into the (f) the Structural descriptor from a leaf-variable $p_k$
Also it would be helpful to indicate where the losses are evaluated.
"Next Frame" should probably be "Reconstruction" (though "Prediction" might be a more accurate word).
---

Section 4.2:
Notational point, it seems k can be from 1 to d.  But in Section 3 you say it can be from 1 to "n". Maybe it would be clearer to change both ("n" and "d") to "K" to emphasize that "k" is an index which goes up to "K".  (edit... now after reading 5.1: Latent Representation, I understand.  If there are n parts in the dataset you use d>n dimensions and the network learns to "drop" the extra ones... it would help to clarify that here).
Structural Descriptor: You say "in practice, we relax the binary constraints"... Yet this is important.. should write down the equation and say how the "soft" version of [i \in S] calculated.
Section 4.3
"elbo" seems like the wrong name for this loss, as it's not an Evidence Lower BOund.  The elbo would be the sum of this loss and the first component of L_{recon}.  It is a regularizing term, so you could call it L_reg.
It's not obvious that sparse local motion maps mean a heirarchical tree structure, but I see how it could help.  I suggest that without evidence for this loss you soften the claim to "this is intended to help encourage the model to learn a heirarchical tree structure"
Figure 4:
It appears that each row indicates a different video in the dataset, but then in 4f you still have two rows but they appear to correspond to different algorithms... a vertical separator here might help show that the rows in 4f do not correspond to the rows in 4a-e.
"Next Frame" here appears to refer to ground truth, but in Figure 3 "Next Frame" appears to be the prediction (which you call reconstruction).
Section 5.1

Future Prediction: No explanation either here or in Figure 4 of what it actually shows.  (What would a failure look like?)
Hierarchical structure... You binarize... how?
Figure 9:
What does this show?  The figure does not obviously demonstrate anything.  Maybe compare to ground-truth future frames?
Section 5.3:
Future Prediction: These images are from the test set, right?  If so, that is worth mentioning.
Object Segmentation ("in several different dimensions"  -> "corresponding to the active latent dimensions?")
Object Segmentation: Visually, it looks nice, but I have now idea how good this segmentation is.  You compare verbally to R-NEM and PSD, but there're no numbers.
Human Studies... The line "For those whose predicted tree structures are not consistent with ours, they all agree with our results and believe ours are more reasonable than others" .. brings to mind an image of a room full of tortured experimental subjects not being allowed to leave until they sign a confession that their own tree structures were foolish mistakes and your tree structures are far superior.... So probably it should be removed because it sounds shady.

---

> ### Author Response · Authors · 2018-11-16
> **Our Response to Reviewer 4**
>
> Thank you very much for the constructive comments. We respond to the major comments below. We’ll also update figures, rewrite captions, and clarify notations in our revision.
>
> 1. Dimensionality of the latent variables
> We agree that motion has more than one degree of freedom; here, the network learns a mapping from 1-D variable to the motion manifold (similar to GANs that learn to map a 100-D variable to the image manifold).
>
> 2. Problem setup and baselines
> Our model can be considered as a conditional VAE, and it behaves differently during training and testing.
>
> 1) During training, we feed the image of current frame I1 and the flow between current and next frame M = flow(I1, I2) into the model as inputs. Our model tries to reconstruct the flow M and leverages it to synthesis the image of next frame I2.
>
> 2) During testing, the model only sees a **single** frame. It samples the latent variable to generate possible motion kernels. It then makes use of the sampled motions to estimate the flow between current and next frame, and leverages the flow to synthesis possible next frames.
>
> During training, we choose not to feed the input flow directly into the image decoder, because the model will have no access to the flow during testing. We will revise our paper to emphasize the different inputs we are using during the training and testing time (in Figure 3).
>
> Our goal is to learn the prior that ties part structure and dynamics to their appearance. Only with the learned prior, our model can segment object parts and synthesize motion from a single image. For example, the “torso” is always the parent of the “leg”, and the motion of the leg is always affected by the motion of the torso. Therefore, in our datasets, parts always have the same hierarchical tree structure.
>
> We agree with the reviewers that it’d be important to add more baselines on future prediction. Note that our model takes a single image for segmentation and future prediction, while most baselines require multi-frame input (e.g., requiring the previous motion field). We will make a comparison with 3DcVAE [1], which only needs one frame as input. For object segmentation, we have included quantitative results on shapes and digits, and will also include numbers for humans in the revision.
>
> 3. Structural descriptor
> We study the problem where object parts share the same hierarchical structure (e.g. ‘leg’ is always part of ‘full torso’). Therefore in our framework, the structural matrix S is shared across data points. In Equation 3, the binary indicator $[i \in P_k]$ represents whether $O_i$ is an ancestor of $O_k$, and we replace this binary indicator with a continuous value $S_{ik}$ to make the entire framework differentiable: $S_{ik} = sigmoid(W_{ik})$, where $W_{ik}$ are trainable parameters. Then, we make use of these relaxed indicators $S_{ik}$ to combine the motion maps using Equation 1-3. During evaluation, we binarize the values of $S_{ik}$ by a threshold of 0.5 to obtain the hierarchical tree structure. We’ll include this in the revision.
>
> 4. Human studies
> Thanks for the suggestion. We agree that the experiment is not well explained and will remove it in the revision.
>
> We have also listed all other planned changes in our general response above. Please don’t hesitate to let us know for any additional comments on the paper or on the planned changes.
>
> Reference:
> [1] Li, Yijun and Fang, Chen and Yang, Jimei and Wang, Zhaowen and Lu, Xin and Yang, Ming-Hsuan. Flow-Grounded Spatial-Temporal Video Prediction from Still Images. In ECCV, 2018.

---

> > ### Comment · AnonReviewer4 · 2018-11-19
> > **Answer**
> >
> > (1) The example you give with GANs mapping 100D onto image space doesn't really compare to this.  There, it is assumed that images lie on a "true" 100D manifold.  Here, you know that the "true" manifold of motion is >1D, so using a 1D latent variable seems like an odd choice.
> >
> > (2) Thank you for clarifying how the test-time situation differs. I think the paper would benefit greatly from an "Algorithms" box, where you explicitly spell out the training and test time performance, and, e.g. how the structural descriptor is calculated.

---

> > > ### Author Response · Authors · 2018-11-27
> > > **Our Response to Reviewer 4**
> > >
> > > Thank you very much for the comments.
> > >
> > > (1) We agree that the true manifold of motion is 2D. Here, we focus on learning the conditional motion distribution: for a particular segment (e.g. left arm), its possible motion distribution in the training set does not encompass all possible 2D motions. As you suggested, we’re assuming that the set of conditional motions lie on a ‘true’ 1D manifold.
> > >
> > > (2) Thanks for the valuable suggestion. We’ve added an algorithm box in the revision to demonstrate the training and evaluation setup.
> > >
> > > The general response above summarized the other changes we’ve made in the revision. Thanks again for your comments, and please don’t hesitate to let us know if you have additional feedback.

---

> > > > ### Author Response · Authors · 2018-12-04
> > > > **Our Response and Revision**
> > > >
> > > > Dear Reviewer 4,
> > > >
> > > > Thanks again for your constructive reviews, which have helped us improved the quality and clarity of the paper. In particular, in the revision, we have included an algorithm box, revised the method section, and updated the figure to better explain the algorithm and its setup. We have also included a systematic comparison with the state of the art on both structure recovery (R-NEM) and future prediction (3DcVAE).
> > > >
> > > > As the discussion period is about to end, please don’t hesitate to let us know if there are any additional clarifications that we can offer, as we would love to convince you of the merits of the paper. We appreciate your suggestions. Thanks!

---

### Public Comment · ~Sjoerd_van_Steenkiste1 · 2018-11-06
**R-NEM / RNN-EM being unable to deal with occlusion**

Hi,

I am the first author of the R-NEM paper. I read your paper and was very impressed with the results.

One thing I was confused about is the performance of R-NEM / RNN-EM on the shapes and digits task (figure 7). You report that the performance is poor because "... [R-NEM / RNN-EM] cannot deal with highly occluded objects". However, this claim does not match the results from our own experiments, eg. on the flying shapes task [1] in Figure 4 one can see 5 object simultaneously occluding one another, or on the bouncing balls with curtain task [2] an invisible curtain occludes the balls.

This leads me to wonder how you train R-NEM / RNN-EM in your experiment. Since your approach trains on pairs of (x_t, flow(x_{t-1}, x_t)) -> (x_{t+1}), could it be that you are only using sequences of length T=2 time-steps to train R-NEM / RNN-EM?

If this is the case then that would explain its poor performance. R-NEM / RNN-EM rely on iterative inference, requiring several steps (each approximately corresponding to an EM step) to obtain good masks from the random initial masks. In our experiments on videos we have always opted to take 1 EM step per time-step as we considered long sequences (>20 steps), which would ensure convergence.  An example of this convergence behavior can be seen in the first couple of steps in Figure 4 in [1].

Cheers,

Sjoerd

[1] Greff, K., van Steenkiste, S., & Schmidhuber, J. (2017). Neural expectation maximization. In Advances in Neural Information Processing Systems (pp. 6691-6701).
[2] van Steenkiste, S., Chang, M., Greff, K., & Schmidhuber, J. (2018). Relational neural expectation maximization: Unsupervised discovery of objects and their interactions. (2018). International Conference on Learning Representations.

---

> ### Author Response · Authors · 2018-11-16
> **Our Response**
>
> Thank you very much for the comments.
>
> In our experiments, we trained the R-NEM / RNN-EM on sequences of 20 frames, where the two input frames appear repetitively: (I1, I2, I1, I2, …, I1, I2). We found that using only two frames is not sufficient because of the exact reason as you mentioned. For evaluation, we still feed 20-frame sequences into the R-NEM / RNN-EM, while our PSD model only takes a single frame as input. In the revision, we will include results on a new dataset of longer video sequences. Thanks again for your suggestions.
>
> We feel R-NEM / RNN-EM and our PSD model focus on complementary topics: R-NEM / RNN-EM learns to identify instances through temporal reasoning, using signals across the entire video to group pixels into objects; our PSD model learns the appearance prior of objects: by watching how they move, it learns to recognize how object parts can be grouped based on their appearance and can be applied on static images. An interesting future work is to explore how these models can be integrated.

---

### Author Response · Authors · 2018-11-16
**Our General Response**

We thank all reviewers for their comments. In addition to the specific response below, here we summarize our task, setup, and the changes planned to be included in the revision.

Our task is to simultaneously learn, without annotation,
1) segmenting object parts;
2) the hierarchical structure of low-level concepts;
3) their dynamics for future prediction.

While our model sees two frames during training, it takes only a **single** image as input during testing. It segments object parts in the image, infers their structure, and synthesizes multiple future frames based on the inferred segments, structure, and the sampled motion vector z. The task is highly non-trivial, as it requires the model to associate object appearance with their possible motion purely from unannotated data.

Our model differs from pure video prediction algorithms, which do not model object structure or relation. Only very recently, researchers have started to build models for the same purpose (RNEM). While RNEM works on synthetic binary images, our model performs well on real color images that are much more complex.

Following the reviews, we plan to include the following changes in the revision by Nov. 26 (the new official revision deadline, extended from Nov. 23)
1) We will cite and discuss the suggested related work.
2) We will revise the method section and the figures for better clarity.
3) While future prediction is not our focus, we agree with the reviewers that it’d be important to add more baselines. Note that our model takes a single image for future prediction, while most video prediction algorithms require multi-frame input. We will make a comparison with 3DcVAE [1] which only needs one frame as input.
4) We will include quantitative results on unsupervised part segmentation on real images, in addition to the results on shapes and digits.
5) We will include the results of RNEM on longer video sequences, as suggested in the public comment.

Please don’t hesitate to let us know for any additional comments on the paper or on the planned changes.

Reference:
[1] Li, Yijun and Fang, Chen and Yang, Jimei and Wang, Zhaowen and Lu, Xin and Yang, Ming-Hsuan. Flow-Grounded Spatial-Temporal Video Prediction from Still Images. In ECCV, 2018.

---

### Author Response · Authors · 2018-11-27
**Our General Response**

We thank all reviewers for their comments. We have revised our manuscript accordingly. Specific changes include

1) We’ve added an algorithm box to show how the model works in training and testing: during training, it learns from unlabeled, paired frames; during testing, it segments object parts, infers their dynamics, and synthesize multiple possible future frames, all from a single image.

2) We have systematically compared our model with the state of the art (R-NEM) on this very challenging task. While R-NEM only works on binary images of shapes and digits, our model works well on real images of complex texture and background (Section 5.3). We’ve also included a systematic study about R-NEM’s ability to handle occluded objects in Appendix A.5, Figure A3, and Table A1.

3) We’ve included comparison on future prediction with 3DcVAE [1] (Section 5.3 and Figure 11).

4) We’ve added quantitative results on unsupervised human part segmentation on real images in Table 2 and Section 5.3.

5) We’ve revised the method section (Section 3) and the corresponding Figure 3, and rewrote the captions for better clarity.

6) We’ve cited and discussed the suggested related work in Section 2.

---

### Meta-Review · Area_Chair1 · 2018-12-14
**novel method for learning part hierarchies and their motion dynamics**

**Confidence:** 5
**Recommendation:** Accept (Poster)

**Metareview:**

The paper proposes a novel method that learns decompositions of an image over parts, their hierarchical structure  and their motion dynamics given temporal image pairs. The problem tackled is of great importance for unsupervised learning from videos. One downside of the paper is the simple datasets used to demonstrate the effectiveness of the method.  All reviewers though agree on it being a valuable contribution for ICLR.

In the related work section the paper mentions "...Some systems emphasize
learning from pixels but without an explicitly object-based representation (Fragkiadaki et al., 2016 ...". The paper you cite in fact emphasized the importance of having object-centric predictive models and the generalization that comes from this design choice, thus, it may be potentially not the right citation.